# Comparing Absorbable and Nonabsorbable Suture Materials for Repair of Achilles Tendon Rupture: A Magnetic Resonance Imaging-Based Study

**DOI:** 10.3390/diagnostics10121085

**Published:** 2020-12-13

**Authors:** Jaeho Cho, Hyun-Joo Kim, Jeong Seok Lee, Jahyung Kim, Sung Hun Won, Young Yi, Dong-Il Chun

**Affiliations:** 1Department of Orthopaedic Surgery, Chuncheon Sacred Heart Hospital, Hallym University, 77, Sakju-ro, Chuncheon-si 24253, Korea; hohotoy@nate.com; 2Department of Radiology, Soonchunhyang University Seoul Hospital, 59, Daesagwan-ro, Yongsan-gu, Seoul 04401, Korea; reonora@schmc.ac.kr; 3Department of Orthopaedic Surgery, Soonchunhyang University Seoul Hospital, 59, Daesagwan-ro, Yongsan-gu, Seoul 04401, Korea; 124856@schmc.ac.kr (J.S.L.); hpsyndrome@naver.com (J.K.); orthowon@gmail.com (S.H.W.); 4Department of Orthopaedic Surgery, Seoul Foot and Ankle Center, Inje University Seoul Paik Hospital, 85, 2-ga, Jeo-dong, Jung-gu, Seoul 04551, Korea; 20vvin@naver.com

**Keywords:** Achilles tendon rupture, magnetic resonance imaging, suture material, tendon healing

## Abstract

We aimed to compare magnetic resonance imaging (MRI) findings and corresponding clinical outcomes of repaired Achilles tendons using absorbable and nonabsorbable sutures. Patients who underwent Achilles tendon repair were divided into 2 groups, with 11 in the absorbable group (group A) and 11 in the nonabsorbable group (group B). For all patients, MRI findings taken 6 months postoperatively were evaluated for morphological changes in the tendon. Concurrently, the American Orthopedic Foot and Ankle Society (AOFAS) ankle-hind foot score and incidence of postoperative complications were evaluated. Regarding MRI findings, the extent to which the cross-sectional area of the repaired tendon was thicker than that of the preoperative tendon was significantly greater in group B than in group A (*p* = 0.0012). Notably, more stitches remained within the tendon in group B than in group A (*p* = 0.0063). No other MRI findings showed a significant difference between the two groups. No significant difference was observed in the AOFAS score, and there was one re-rupture each in both groups. Because nonabsorbable suture material in the treatment of Achilles tendon rupture yielded a thicker postoperative MRI cross-sectional area, enhanced rehabilitation is recommended in order to prevent scar formation.

## 1. Introduction

The primary goals of the management of acute Achilles tendon ruptures are to promote the permanent tendon healing at the correct length and to ensure a rapid return to normal function. Operative repair of a ruptured Achilles tendon is considered to fulfill these goals, resulting in satisfactory clinical outcomes while simultaneously minimizing the re-rupture rate [1,2]. Among the various issues associated with operative procedures, the selection of the suture material for Achilles tendon repair depends on the surgeon’s preference rather than sufficient evidence supporting an established superior material.

Traditionally, many surgeons are in favor of using nonabsorbable, multifilament sutures, primarily because of the belief that the suture material remains within the repaired tendon and provides adequate fixation strength during the critical healing period [3]. However, because all operative suture materials within the body may cause some degree of inflammatory reaction, extrinsic scar tissue formation, chronic inflammation, and infection, these may affect postoperative clinical outcomes [4,5,6,7].

Considering these issues, it is known that absorbable suture material not only has sufficient holding capacity and strength, but also results in similar postoperative functional outcomes to nonabsorbable suture material [8,9]. Moreover, fewer postoperative complications are observed with absorbable sutures [10,11]. However, to our knowledge, little is known about whether the morphology of the repaired Achilles tendon varies according to suture material because it is not apparent on routinely obtained plain radiographs. Consequently, acknowledgment of the different features of the repaired tendon according to suture material may play a role not only in selecting the adequate suture material, but also in verifying the incidence of postoperative complications among nonabsorbable sutures.

Therefore, the aim of this study was to compare magnetic resonance imaging (MRI) findings and corresponding clinical outcomes of repaired Achilles tendons using absorbable versus nonabsorbable sutures. We hypothesized that tendons repaired with nonabsorbable suture material would become thicker, probably due to more scar formation.

## 2. Materials and Methods

### 2.1. Patients

This prospectively collected, retrospective study was approved by the Institutional Review Board of Soonchunhyang University Seoul Hospital (Institutional Review Board number: SCHUH 2020-07-013), and written informed consent for publication of this report was obtained from all included patients. From July 2017 to December 2019, a total of 60 patients who presented with acute isolated rupture of the Achilles tendon and then received open Achilles tendon repair by a single senior orthopedic surgeon were enrolled. The inclusion criteria were as follows: (1) Isolated, indirect Achilles tendon rupture; (2) minimum follow-up of 6 months after surgery; and (3) presence of postoperative MRI around the 6-month follow-up period. The exclusion criteria were as follows: (1) Open tendinous rupture or laceration; (2) direct, traumatic blunt rupture; (3) subacute or chronic rupture with delayed diagnosis of over 1 week after the injury; (4) previous injury to the same tendon; (5) functional deficit on the contralateral ankle; (6) history of systemic diseases, including neurovascular disease or immune-suppressed disease; and (7) execution of adjunctive procedures, such as flexor hallucis longus transfer or gastrocnemius augmentation. Consequently, 24 patients did not meet inclusion criteria and 6 patients declined to participate. Therefore, 30 patients were not included.

The remaining patients were allocated into 2 groups in terms of the suture materials used to perform the core suture of the ruptured Achilles tendon according to their hospital registration numbers. Patients with odd registration numbers were allocated into group A and were repaired with absorbable braided polyglactin sutures (Vicryl^®^, ETHICON, Johnson & Johnson, Seoul, Korea), while those with even registration numbers were allocated into group B and were repaired with nonabsorbable braided polyethylene terephthalate sutures (Ethibond^®^, ETHICON, Johnson & Johnson, Seoul, Korea). Postoperatively, 8 patients were excluded and data from the remaining 22 patients were included for further analysis (Figure 1).

### 2.2. Operative Technique

All procedures were performed with the patients in a prone position under regional or general anesthesia with a tourniquet applied at the thigh. A 5 cm longitudinal skin incision was made medial to the tendon, and the paratenon and mesotenon were incised longitudinally and retracted to expose the ruptured tendon. The ruptured tendon was repaired using a 2-stranded double Krackow suture technique. In terms of suture material used for the core suture, No. 2 Vicryl^®^ was used in group A, while No. 2 Ethibond^®^ was used in group B. Following the core suture, the augmented tendon was supplemented with interrupted circumferential sutures and the paratenon was closed in both groups using 3-0 Vicryl^®^.

### 2.3. Postoperative Management

The same rehabilitation protocol was applied to all 22 patients in this study. The operated ankle was immobilized with a short leg cast for 2 weeks in a non-weight-bearing position of natural plantar flexion. Two weeks postoperatively, tolerable weight-bearing in a functional brace was initiated and ankle joint motion from full plantar flexion to 20° dorsiflexion was allowed. The dorsiflexion angle of the ankle joint increased gradually every week. Six weeks postoperatively, the patients were allowed to bear full weight as tolerated using a functional brace. Additionally, the ankle joint was permitted the full range of motion along with the strengthening and distracting exercise of the muscles around the Achilles tendon using a rubber band. From 8 weeks postoperatively, patients were allowed to walk with normal shoes. Three months after surgery, sports activity, including running, was allowed. At the 6-month postoperative follow-up, in addition to a postoperative MRI, patients were asked to complete the American Orthopedic Foot and Ankle Society (AOFAS) ankle-hindfoot score [12]. These scores were collected and analyzed retrospectively, together with the incidence of postoperative complications through medical record reviewing.

### 2.4. Imaging Protocol

For all pre- and postoperative ankle MRI examinations, 1.5-T (Sonata, Siemens Healthineers, Erlangen, Germany) and 3-T (Discovery MR750w, GE Healthcare, Milwaukee, WI, USA) MRI scanners were used with a commercially available ankle coil. The MR scanning protocols are shown in Table 1. Sagittal T1-weighted and fat-saturated T2-weighted images and axial fat-saturated T2-weighted images were obtained for postoperative ankle MRI evaluations. During MRI, the patient remained in supine with the affected Achilles tendon placed on the coil. The affected leg was positioned so that the ruptured site on the Achilles tendon was covered on the sagittal scout-view image.

### 2.5. Assessment

Preoperative MRI was performed a day before the surgery, while postoperative MRI was performed around 6 months after surgery. Images were independently evaluated by 2 fellowship-trained orthopedic surgeons who had 10 and 5 years of experience, respectively, in musculoskeletal imaging. Then, the final decision of the MRI finding was made based on the consensus between the 2 surgeons through detailed discussion. The following parameters to evaluate the integrity of the repaired tendon via MRI were based on the results of previous studies [13,14,15].

#### 2.5.1. T1-Weighted Sagittal Image

On the T1-weighted sagittal image, the postoperative appearance of the thickened Achilles tendon and the remaining stitches within the tendon were evaluated. The postoperative appearance of the thickened Achilles tendon was classified into 3 categories: (1) Diffusely thickened with an isointense signal, (2) diffusely thickened with a hyperintense signal, and (3) focally thickened into a fusiform appearance with a diffuse hyperintense signal (Figure 2). In addition, the presence of remaining stitches within the tendon was classified into 3 categories in the postoperative MRI through observation of the dark signal intensity artifact: (1) None; (2) mild, with the presence of dark signal intensity artifacts that are not obvious; and (3) marked as the presence of definite signal intensity artifacts (Figure 3) [16].

#### 2.5.2. Fat-Saturated T2-Weighted Axial Image

On the fat-saturated T2-weighted axial image, the postoperative shape of the musculotendinous junction, postoperative signal changes around the tendon, and cross-sectional area changes of the repaired tendon were evaluated. The postoperative shape of the musculotendinous junction was classified into 3 categories: (1) Concave (Figure 4A), (2) flat (Figure 4B), and (3) convex (Figure 4C). The postoperative signal changes around the tendon were also categorized according to location: (1) Circumferentially surrounding the tendon (Figure 4A), (2) focally within the tendon (Figure 4B), and (3) diffusely throughout the tendon (Figure 4C). Finally, the cross-sectional area of the repaired tendon was measured at the thickest portion of the tendon near the rupture site on the postoperative MRI scans (Figure 5A) and on the preoperative sagittal scans (Figure 5B). The change in the cross-sectional area of the tendon after the surgery was evaluated by dividing the postoperative cross-sectional area (Figure 5A) by the preoperative cross-sectional area (Figure 5B).

### 2.6. Statistical Analysis

For all variables, the Shapiro–Wilk normality test showed no evidence of a non-normal distribution. For comparison between the nonabsorbable and absorbable groups, a two-sample *t*-test and Fisher’s exact test were used. A *p* value <0.05 was considered a significant difference. Data processing and statistical analyses were performed using R version 3.3.1 (Foundation for Statistical Computing, Vienna, Austria).

## 3. Results

A total of 22 patients were included in this study, of which 11 patients had absorbable suture material (group A, 50%) for their core suture and 11 had nonabsorbable suture material (group B, 50%). With the numbers available, there was no significant difference between the two groups in terms of age (group A, 41.73 ± 13.7 years; group B, 40.18 ±10.26 years; *p* = 0.768), gender (*p* > 0.99), or body mass index (group A, 26.11 ± 2.22 kg/m^2^; group B, 25.75 ± 3.16 kg/m^2^; *p* = 0.7601). The mean interval from the date of operative repair to postoperative MRI was 177 days for group A and 181 days for group B (Table 2).

With regards to MRI findings, there were statistically significant differences in the cross-sectional area change ratio between the two groups (*p* = 0.0012). The average preoperative cross-sectional area of the ruptured tendon was 1.23 cm^2^ in group A and 1.14 cm^2^ in group B. The average postoperative cross-sectional area of the repaired tendon was 3.01 cm^2^ in group A and 3.64 cm^2^ in group B. As a result, the postoperative cross-sectional area of the repaired tendon became 3.19 ± 0.51-times thicker than that of the preoperative tendon in group B, compared with 2.43 ± 0.41-times thicker in group A. In addition, markedly more stitches remained within the tendon in group B than in group A (*p* = 0.0063). However, with the numbers available, there was no significant difference between the two groups in T2-weighted axial signal intensity change, postoperative appearance, or shape of the musculotendinous junction (Table 3).

In terms of clinical outcomes, no significant difference was observed between the two groups in postoperative AOFAS ankle-hindfoot score (group A, 92.27 ± 7.48; group B, 88.18 ± 8.58; *p* = 0.2475). In addition, there were two re-ruptures recorded at the 6-month follow-up: One in group A and the other in group B. Both re-ruptures occurred in patients who did not follow the rehabilitation protocol provided. These patients both initiated sports activity right after they were allowed to walk with normal shoes at 2 months postoperatively. No other complications were observed (Table 4).

## 4. Discussion

MRI is a useful diagnostic imaging modality for evaluating the integrity of a repaired tendon and detecting the potential causes of postoperative complications [17]. Considering these advantages, previous studies have evaluated the long-term MRI findings of the postoperative healing process after Achilles tendon repair [13,14,15]. In this study, the postoperative MRI findings of Achilles tendons repaired with different suture materials were compared. We found that changes in the cross-sectional area of the operated tendon were significantly different between the two groups. Our hypothesis that tendons repaired with nonabsorbable suture material would become thicker, probably due to more scar formation, was verified.

The healing process of the injured tendon is known to comprise a 1-year course of serial, overlapping stages, namely an inflammatory stage, a proliferative stage, and a remodeling stage. Following 6 weeks of phagocytosis and collagen synthesis, the remodeling phase begins, which can be divided into the consolidation and maturation stages. During the consolidation stage, the repaired tissue transforms from cellular to fibrous, and collagen fibers become aligned in the direction of stress. Finally, tendon healing reaches the maturation stage at which point the fibrous tissue gradually becomes converted to scar-like tissue, followed by a declination of tendon vascularity and tenocyte metabolism [18].

This complicated tendon healing process may take place either intrinsically, by epi- and endotenon tenocyte proliferation, or extrinsically, by the migration of tenocytes from the surrounding sheath and synovium [19,20,21]. To distinguish between the two, intrinsic healing produces a normal gliding mechanism within the tendon sheath along with fewer complications, while extrinsic healing results in scar formation, adhesion with adjacent tissue, and disruption of the gliding mechanism [22,23]. The relative contribution of intrinsic and extrinsic tendon healing may be influenced by the type of initial trauma, anatomical location, existence of a synovial sheath, and amount of motion-induced stress after the operative repair [24]. In fact, tendons with low cellularity and vascularity are especially prone to extrinsic tendon healing, with the deposition of excessive disorganized extracellular matrices [25,26].

In this study, compared to tendons repaired using absorbable suture material, those repaired with nonabsorbable suture material clearly demonstrated thicker, hypointense fibrous scar tissue. Although the resulting scar provides some level of tissue stability, the mechanical integrity of the original tendon is altered [27]. In line with scar formation within the repaired tendon, intratendinous fascicle sliding becomes impaired, and it may have a negative effect on force transmission during ankle motion. In addition, tendon function can be modified during stretch-shortening exercise and can be assumed to decrease tendon elasticity. As a result, modulation of tendon function is produced, which may eventually result in an elevated risk of tendon re-rupture [28]. Furthermore, fibrotic changes to the tendon extracellular matrix after the injury are thought to contribute to the subsequent development of chronic, degenerative tendinopathies [29]. Therefore, enhanced rehabilitation to promote intrinsic tendon healing, minimize extrinsic scar tissue formation, optimize tendon gliding, and restore functional outcome are recommended in tendons repaired with nonabsorbable suture materials [30,31,32].

In spite of such differences in scar tissue formation between distinct suture materials, no significant difference was found between the two groups in terms of AOFAS ankle-hindfoot score or postoperative complications. Similarly, previous studies have shown that absorbable and nonabsorbable suture materials have similar postoperative functional outcomes, although more postoperative complications have been observed with nonabsorbable sutures [5,10,11]. Nevertheless, it should not be overlooked that the tendons appearing in MRI in this study were still in the process of healing, as all of them were taken approximately 6 months postoperatively. In other words, a longer follow-up period is necessary to accurately correlate the tendon image findings with clinical outcomes. Furthermore, it would be appropriate to approach the image findings carefully, because an abnormal image does not necessarily imply clinically relevant pathology or poor function [33].

Despite absorbable stitches disappearing within an average of less than 3 months, not all absorbable stitches disappear from the tendon by the 6-month postoperative period [34]. In other words, the potential risk of a foreign body reaction remains in absorbable suture materials. In addition, previous studies have reported that the tendon becomes elongated following operative repair, which is associated with inferior clinical outcomes [35]. In the same manner, we also tried to identify the amount of tendon elongation after surgery according to suture material. However, the lack of a control group made it difficult to estimate the pre- to postoperative difference, because MRI of the contralateral leg could not be performed due to its high cost and limitations of the coil design. Accordingly, issues such as calculating the actual absorption timing of the absorbable suture material within the human Achilles tendon over a longer follow-up period, as well as comparing distinct amounts of tendon elongation depending on different suture materials, represent a scope for future study.

This study is limited due to its nonrandomized, retrospective nature with a small sample size of 22 cases with short-term follow-up. In addition, a priori sample size calculation was not performed, owing to the small number of included patients in this study. Not all patients agreed to additional MRI during the postoperative follow-up period due to the relatively high cost of MRI. A follow-up study that accumulates sufficient cases would strengthen the validity of this study. Further, only single types of nonabsorbable and absorbable suture materials were used in this study. However, Ethibond^®^ and Vicryl^®^ are the most commonly used suture materials because of their high tension strength and minimal tissue reaction. Therefore, we believe that the results from the present study do provide valuable information for orthopedic surgeons despite the limited types of suture material evaluated. Finally, no additional diagnostic modality other than MRI was used to evaluate the integrity of the Achilles tendon, which weakens the evidence of the present study. Tissue biopsy of the thickened portion of the tendon seen on MR images at the postoperative 6-month period might clearly explain the histologic difference of the thickened tendon upon different suture materials. However, such invasive procedures were ethically unacceptable.

## 5. Conclusions

In conclusion, the use of a nonabsorbable suture material in the treatment of Achilles tendon rupture demonstrated a thicker postoperative cross-sectional area on MRI scans. Although a careful approach would be required since this is a short-term follow-up study, we recommend an enhanced rehabilitation protocol in tendons repaired with nonabsorbable suture material in order to prevent excessive scar formation.

## Figures and Tables

**Figure 1 diagnostics-10-01085-f001:**
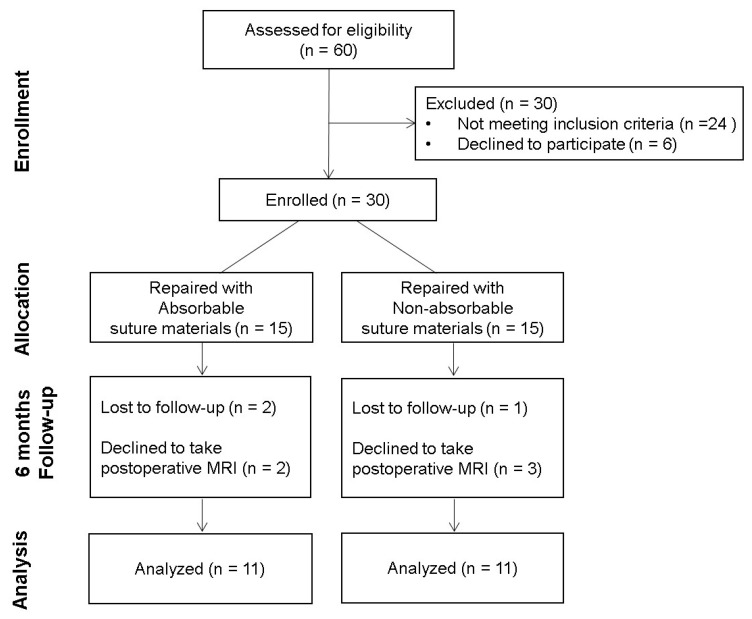
Flow diagram of the included patients. MRI = magnetic resonance imaging.

**Figure 2 diagnostics-10-01085-f002:**
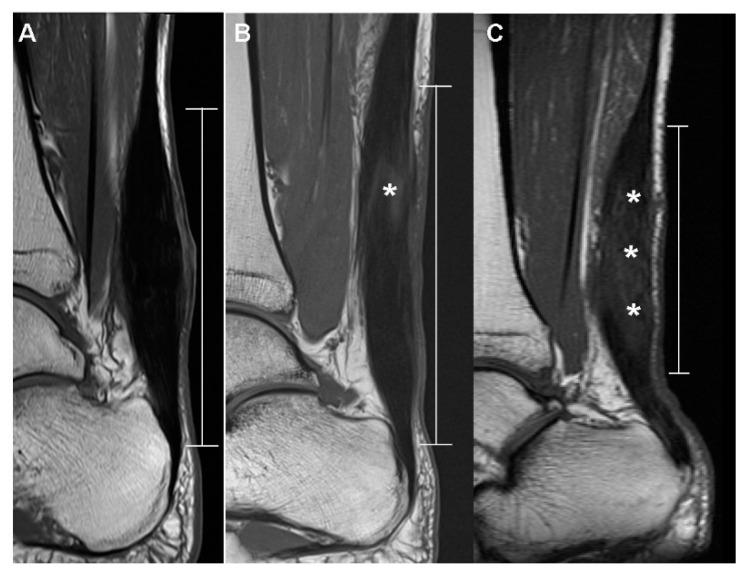
Postoperative appearance of the thickened Achilles tendon on the T1-weighted sagittal image. Thickened portion of the tendon is indicated as the interval between the normal portions of the tendon. Each image is representative subject classified into 3 categories: (**A**) Diffusely thickened with an isointense signal, (**B**) diffusely thickened with a focal hyperintense signal (asterisk), and (**C**) focally thickened into a fusiform appearance with a diffuse hyperintense signal (asterisks).

**Figure 3 diagnostics-10-01085-f003:**
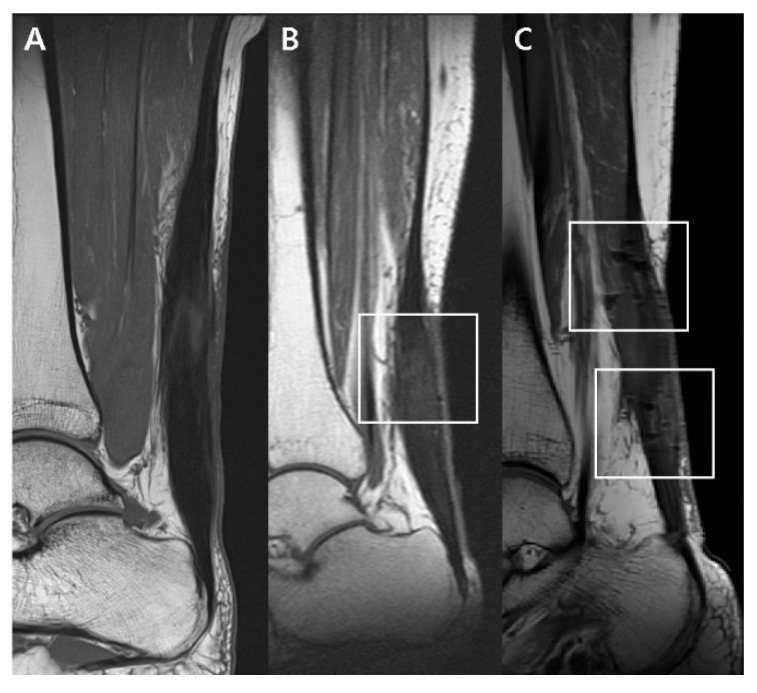
Presence of remaining stitches within the tendon on the T1-weighted sagittal image through observation of the dark signal intensity artifact (within the boxes). Each image is representative subject classified into 3 categories: None (**A**), mild (**B**), and marked (**C**).

**Figure 4 diagnostics-10-01085-f004:**
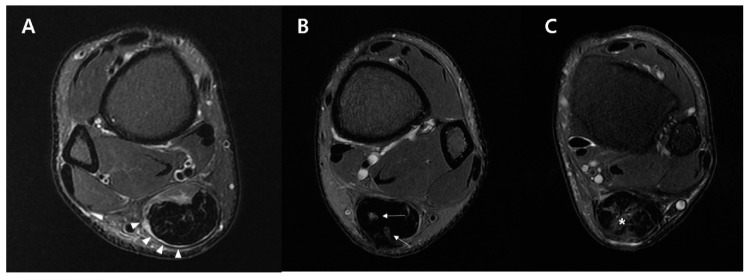
Postoperative changes of the tendon on the fat-saturated T2 weighted axial image. Each image is representative subject classified into 3 categories. Shape of the musculotendinous junction: Convex (**A**), flat (**B**), and concave (**C**). Signal changes around the tendon upon their location: Circumferentially surrounding the tendon (**A**, arrowheads), focally within the tendon (**B**, arrows), and diffusely throughout the tendon (**C**, asterisk).

**Figure 5 diagnostics-10-01085-f005:**
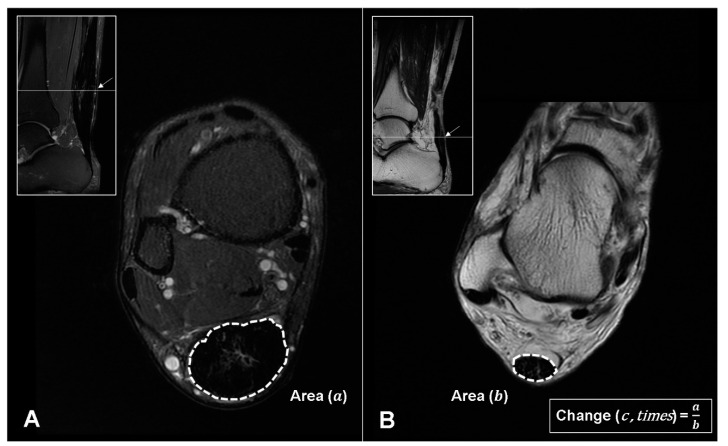
(**A**) Postoperative cross-sectional area (cm^2^) of the repaired tendon (a) measured at the thickest portion of the tendon (arrow) near the rupture site. (**B**) Preoperative cross-sectional area (cm^2^) of the ruptured tendon (b) measured at the thickest portion of the tendon (arrow) near the rupture site. Postoperative changes (c) were then compared by diving the two measured parameters (c = a/b).

**Table 1 diagnostics-10-01085-t001:** MR scanning protocol for postoperative ankle MRI evaluation.

	*Sag T1* *(1.5T)*	*Fat-Sat Sag T2* *(1.5T)*	*Fat-Sat Ax T2* *(1.5T)*	*Sag T1* *(3T)*	*Fat-Sat Sag T2* *(3T)*	*Fat-Sat Ax T2* *(3T)*
**Repetition time (msec)**	490	3000	4720	582	3489	5557
**Echo time (msec)**	12	81	97	7.30	78.4	82.76
**FOV (cm)**	220	220	160	200	200	140
**Matrix size**	384 × 176	384 × 204	320 × 163	416 × 288	416 × 288	384 × 256
**Slice thickness/interval (mm)**	3/0.3	3/0.3	5/1.5	3/0	3/0	4/0.8
**ETL**	1	9	11	4	14	12
**NEX**	2	2	3	2	3	3

MR = magnetic resonance, Sag = sagittal, Fat-sat = fat-saturated, ax = axial, FOV = field of view, ETL = echo train length, NEX = number of excitations.

**Table 2 diagnostics-10-01085-t002:** Demographics.

	Absorbable (A)	Nonabsorbable (B)
Number of cases	11	11
Age (years)	41.73 ± 13.7	40.18 ± 10.26
Gender		
Male	9 (81.82%)	9 (81.82%)
Female	2 (18.18%)	2 (18.18%)
BMI (Kg/m^2^)	26.11 ± 2.22	25.75 ± 3.16
MRI Interval (Days)	177	181

BMI = body mass index, MRI = magnetic resonance imaging.

**Table 3 diagnostics-10-01085-t003:** MRI findings.

	Absorbable (A)	Nonabsorbable (B)	*p*-Value
**T1 sagittal image**			
** Postop appearance**			0.0789
** Diffusely isointense thickened**	5 (45.45%)	1 (9.1%)	
** Diffusely elongated**	3 (27.27%)	2 (18.2%)	
** Focally fusiform**	3 (27.27%)	8 (72.7%)	
** Dark SI artifact**			0.0063
** None**	3 (27.27%)	0 (0%)	
** Mild**	8 (72.73%)	5 (45.5%)	
** Marked**	0 (0%)	6 (54.5%)	
**Fat-saturated T2 axial image**			
** MTJ shape**			0.8204
** Convex**	4 (36.36%)	6 (54.5%)	
** Flat**	6 (54.55%)	4 (36.4%)	
** Concave**	1 (9.09%)	1 (9.1%)	
** Tendon signal changes**			0.7249
** Circumferential**	6 (54.55%)	5 (45.4%)	
** Focal**	3 (27.27%)	3 (27.3%)	
** Diffuse**	2 (18.18%)	3 (27.3%)	
** Postop circumferential area changes (times)**	2.43 ± 0.41	3.19 ± 0.51	0.0012

MRI = magnetic resonance image, SI = signal intensity, MTJ = musculotendinous junction.

**Table 4 diagnostics-10-01085-t004:** Postoperative clinical outcomes.

	Absorbable (A)	Nonabsorbable (B)	*p*-Value
**AOFAS ankle-hindfoot score**	92.27 ± 7.48	88.18 ± 8.58	0.2475
**Complications**			
** Re-rupture, *n***	1	1	
** Infection, *n***	0	0	
** Foreign body reaction, *n***	0	0	

AOFAS = American Orthopaedic Foot and Ankle Society.

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
