# Peer review of "Comparing Absorbable and Nonabsorbable Suture Materials for Repair of Achilles Tendon Rupture: A Magnetic Resonance Imaging-Based Study"

_diagnostics, 2020, doi:10.3390/diagnostics10121085_

Round 1
Reviewer 1 Report
As I mentioned, all surgeons (from trauma and orthopedic surgeons; foot and ankle specialists, as well as plastic surgeons) may benefit from the results - MRI can be a valuable diagnostic tool for eg. adjusting the rehabilitation protocol according to the provided images, etc
Author Response
As I mentioned, all surgeons (from trauma and orthopedic surgeons; foot and ankle specialists, as well as plastic surgeons) may benefit from the results - MRI can be a valuable diagnostic tool for eg. adjusting the rehabilitation protocol according to the provided images, etc
-> Thank you for your generous comment toward our manuscript, and we appreciate it.
Reviewer 2 Report
December 05, 2020
Journal: Diagnostics
ID Number: diagnostics-1029123
Title: Comparing Absorbable and Non-Absorbable Suture Materials for Repair of Achilles Tendon Rupture: A Magnetic Resonance Imaging-Based Study
General comments:
In general, the study was to explore clinical findings of the patients, who underwent isolated-ruptured Achilles Tendon treatment repaired by a senior orthopedic surgeon and were also separated into absorbable (Group A) and non-absorbable (Group B) two suturing groups. Magnetic Resonance Images (MRI) of the sutured tendons were examined and analyzed to determine which suture material (absorbed Vicryl vs. non-absorbed Ethibond) less responding to the body would further provide rehabilitation enhancement needed for the patients by revealing more tendon’s scar formation in terms of thickness and cross-sectional area seen in the images. Although the main portions of the paper are written clearly, there is still unclear information that could be clarified, specifically, for the sections of statistical analysis, results, and discussion.
One of the main unclear information is related to scar formation and stitches remaining of the sutured tendons that were solely determined by the thickness & cross-section area of the repaired tendon and the signal intensity artifact, respectively, given by the MRI result changes examined between pre and post operations. In the study, scar formation was assumed to be positively correlated to the thickness & cross-section area of the repaired tendon, without reporting quantitative thick measurements of the repaired tendons although the cross-section area changes with no unit given were reported; similarly, the stitches remaining of the repaired tendons was also assumed to be correlated to the signal intensity artifacts measured by the MRI. If both assumptions could have well cited with various study references shown in the introduction, the findings of the study will be conclusive and valuable.
There is some other unclear information that needs to be verified as the following contents.
Abstract
The line 24, “… the cross-sectional area of the repaired tendon …was significantly greater in the group A than in the group B (p = 0.0012).” Based on the result, the non-absorbable group B has a greater cross-sectional area than that of group A.
Introduction
The Line 57: “…hypothesized that tendons repaired with absorbable suture material…demonstrate less scar formation.” The relation between an absorbable suture material and tissue scar formation could have been provided with various supportive references so that the hypothesis and later conclusion could be well made.
Materials and methods
The line 86, “All procedures were performed with the patient in the prone position…” The patient should add s to be plural, patients and in “a” prone position.
The line 90, “…No. 2 Ethibond was used in group A, while No. 2 Vicryl was used in group B.” Either brand name of the suture material or group should be corrected according to its absorbability. .
The line 95, “The same rehabilitation protocol was applied to all 20 patients in this study.” The number of the patients is not correctly reported according to 22 patents recruited and analyzed in the study.
The line 105, “…the American Orthopedic Foot and Ankle Society (AOFAS) ankle-hindfoot score.” The purpose of applying the AOFAS score to the patients could have been cited with some references briefly describing for patient’s functional outcome and performance.
The line 124, “…and was based on consensus.” What specific event a consensus was based on?
The line 136, in the Figure 2, “Postoperative appearance of the thickened Achilles tendon…” The thickened area of the tendon could have been indicated with arrows shown on each image of the Figure.
The line 136 & 141, in the Figure 2 & 3, each Figure should have indicated that each image shows as representative subjects classified into 3 categories of each.
The line 141, in the Figure 3, “…observation of the dark signal intensity artifact…” It is very hard to observe the artifact from the figure even though the arrows were drawn on the mild (B) and marked (C).
The line 156, in the Figure 4, the darkened and distorted images of B and C, respectively, are very hard to be viewed.
The line 162, in the Figure 5, the rupture sites indicated with each arrow in the A and B figures look very different relative to the center of the ankle joint.
The line 168, “…was used for continuous variables,…” what specific continuous variables were measured in the study? It should be clarified or removed from this statement.
Results
The line 189, in the Table 3, the first column of the listed categories like Postop appearance, Dark SI artifact, MTJ shape, Tendon signal changes, and Postop circumferential area changes, could have well bolded for a clear view. What is the specific unit of the area changes of the postoperative circumference?
The line 192, “…AOFAS ankle-hindfoot score with the numbers available…” What are the numbers indicated here?
The line 193, “…there were 2 re-ruptures recorded at the 6 month follow-up…” A brief description of under what specific condition re-ruptures of tendons of the patients occurred could have been provided.
Discussion
The line 226, “…the mechanical integrity of the original tendon is altered.” References could have been cited to support the mechanical integrity of the idea.
The line 235, “…enhanced rehabilitation to promote… tendon healing, minimize …scar tissue formation, optimize tendon gliding, and restore functional outcome … in tendons repaired with non-absorbable suture materials [25].” This reference studied in 2007 could have been updated.
The line 268, ”Although histologic examination of the thickened portion of the tendon seen on MR images would have clearly explained the difference in tendon thickness with different suture materials, such invasive procedures were ethically unacceptable.” This sentence is not clear.

Author Response
please see the attachement
